# Reciprocal MIND MELD: Improving Learning From Demonstration via Personalized, Reciprocal Teaching

**Mariah L. Schrum, Erin Hedlund-Botti, Matthew C. Gomoblay**
Institute for Robotics and Intelligent Machines
Georgia Institute of Technology
mschrum3@gatech.edu, ehedlund6@gatech.edu, matthew.gombolay@cc.gatech.edu

**Abstract:** Endowing robots with the ability to learn novel tasks via demonstrations will increase the accessibility of robots for non-expert, non-roboticists. However, research has shown that humans can be poor teachers, making it difficult for robots to effectively learn from humans. If the robot could instruct humans how to provide better demonstrations, then humans might be able to effectively teach a broader range of novel, out-of-distribution tasks. In this work, we introduce Reciprocal MIND MELD, a framework in which the robot learns the way in which a demonstrator is suboptimal and utilizes this information to provide feedback to the demonstrator to improve upon their demonstrations. We additionally develop an Embedding Predictor Network which learns to predict the demonstrator's suboptimality online without the need for optimal labels. In a series of human-subject experiments in a driving simulator domain, we demonstrate that robotic feedback can effectively improve human demonstrations in two dimensions of suboptimality ($p < .001$) and that robotic feedback translates into better learning outcomes for a robotic agent on novel tasks ($p = .045$).

**Keywords:** meta-learning, personalization, imitation learning

## 1 Introduction

When an individual purchases an in-home cleaning robot, the robot will have to be taught many novel tasks over an extended period of time. The user may have to teach the robot how to move dishes from the dishwasher to the proper location in the cabinets or how to wash the windows and take out the trash. Simply pre-programming these tasks may not be an adequate solution as different users may have differing preferences for how their robot should operate. Therefore, to effectively meet the needs of the end-user, the robot must be capable of successfully learning new tasks quickly via demonstration. Prior work has shown that robots learn well via robot-centric (RC) learning from demonstration (LfD), compared to human-centric LfD [1]. In RC LfD, the human demonstrator provides corrective feedback at each timestep while the robot is rolling out its current policy, which helps the robot learn how to recover from mistakes. While RC LfD works well when the demonstrations are high quality, prior work has also shown that humans find RC LfD unintuitive and tend to provide low-quality corrective demonstrations [2, 3, 4, 5]. Such suboptimality, if left uncorrected, is likely to hinder the robot's ability to learn from end-users.

While several approaches have attempted to improve upon a teacher's ability to provide high quality demonstrations via tutorials and videos [6, 7, 8], prior work has primarily focused on correcting for suboptimality after-the-fact rather than directly improving teaching abilities. For example, prior work introduced MIND MELD [9]. MIND MELD meta-learns a personalized embedding describing the way in which a demonstrator is suboptimal in providing feedback in RC LfD via *calibration tasks*. The calibration tasks are a curated, pre-defined set of policy rollouts with known optimal demonstrations and are meant to capture the way in which a demonstrator is suboptimal. However, we hypothesize that correcting for suboptimal demonstrations under-the-hood as MIND MELD does may not be the best long-term strategy because doing so may 1) contribute to end-users' lack of functional understanding, 2) reinforce suboptimal tendencies, and 3) result in poor performance on out-of-distribution tasks and novel robotic platforms [2, 10, 11].

6th Conference on Robot Learning (CoRL 2022), Auckland, New Zealand.

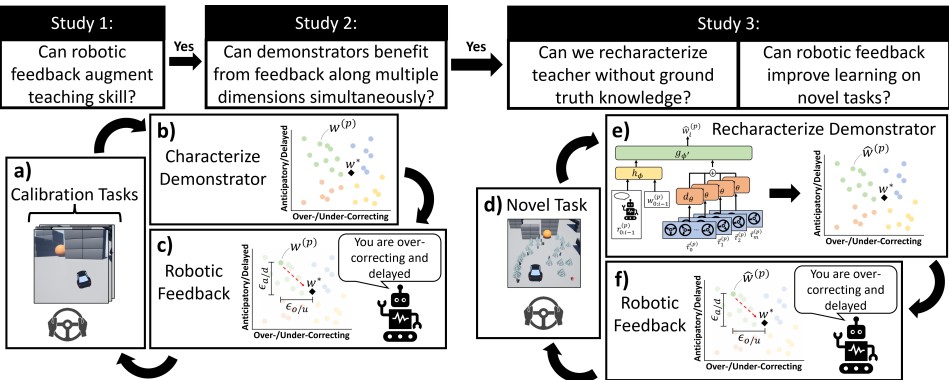

Figure 1: This figure illustrates an overview of our methodology and study designs. Figs 1a, 1b, and 1c show the methodology for Studies 1 and 2 and Figs 1d, 1e, and 1f the methodology for Study 3.

Consequently, there is a need for a framework that can coach demonstrators to become better teachers. To solve this problem, we propose Reciprocal Mutual Information Driven Meta-Learning from Demonstration (Reciprocal MIND MELD). Reciprocal MIND MELD is based upon the MIND MELD framework but is meant to guide the human demonstrator to proactively improve their feedback. Reciprocal MIND MELD differs from MIND MELD in three significant ways. First, Reciprocal MIND MELD learns a *semantically meaningful* personalized embedding via calibration tasks that describes the way in which a demonstrator is suboptimal (Fig. 1b and e). Second, based upon this personalized embedding, Reciprocal MIND MELD provides *robotic feedback* to the demonstrator to improve their teaching abilities (Fig. 1c and f) and consequently improve learning outcomes for the agent rather than correcting for suboptimality retroactively. Third, we introduce an Embedding Predictor Network (EPN) which dynamically updates the demonstrator's personalized embedding by estimating its new location (Fig. 1e), thus eliminating the need to repeat the calibration tasks. In our work, we contribute the following:

1. We propose Reciprocal MIND MELD, a novel method for providing feedback to demonstrators to improve their teaching abilities via a personalized embedding.
2. We develop an EPN to dynamically update the demonstrator's personalized embedding without the need to repeat the time-consuming calibration tasks (Fig. 1e).
3. We demonstrate that Reciprocal MIND MELD can improve an individual's demonstrations ($p < .001$), accurately estimate a demonstrator's new embedding ($p = .002$), and improve learning outcomes of the robot ($p = .045$) in a driving simulator domain.

## 2 Related Work

Researchers are increasingly designing new algorithms to learn from suboptimal demonstrations [12, 13, 14, 15, 16, 17] as well as make LfD more user-friendly [3, 18, 19, 20]. In Chen et al., the authors introduce SSRR which improves upon an agent's ability to learn from suboptimal demonstrations by characterizing the relationship between noise and performance [21]. Brown et al. [16, 22] and Myers et al. [23] improve upon the ability to learn from suboptimal demonstrations by learning a reward function from a ranked set of demonstrations. Schrum et al. introduced MIND MELD [5, 9] which meta-learns a personalized embedding describing a teacher's suboptimal tendencies and was shown to outperform prior work in LfD.

Several approaches have also investigated how best to provide feedback to a demonstrator to improve their demonstrations [6, 7, 8, 24, 25]. Cakmak and Takayama conducted a study investigating several modalities for communicating improvements to a demonstrator. The authors found instructional videos to be the best modality for improving teaching [6]. Sena et al. investigated video feedback with and without rule guidance and found that both modalities produced better results than no feedback [7]. A more extensive discussion of related work can be found in the Appendix.

## 3 Preliminaries

Reciprocal MIND MELD is inspired by the MIND MELD architecture demonstrated in previous work [5]. The objective of MIND MELD is to learn a personalized embedding to describe the way

in which a demonstrator is suboptimal in an RC LfD paradigm, where the demonstrator provides corrective feedback to the robot. MIND MELD then utilizes this embedding to map a demonstrator's suboptimal demonstrations to demonstrations closer to optimal. The MIND MELD architecture (shown in gray in Fig. 2) is trained via *calibration tasks*, which are used to learn the mapping ($f_\theta$, $\mathcal{E}_{\phi'}$, and $q_\phi$) from suboptimal labels, $a_{t-\Delta t:t+\Delta t}^{(p)}$, to better labels, $\hat{d}_t^{(p)}$, and learn the personalized embedding, $w^{(p)}$, representing an individual demonstrator. The calibration tasks consist of a set of pre-recorded policy rollouts with known optimal labels. Participants provide corrective demonstrations to the robot during these rollouts to direct the robot to a goal. MIND MELD learns to map the participant's corrective labels to higher-quality labels while simultaneously inferring the personalized embedding, $w^{(p)}$, representing an individual, $p$'s, suboptimal style. To ensure that $w^{(p)}$ can represent various and distinct feedback styles, MIND MELD maximizes a lower bound on mutual information between the way in which a demonstrator is suboptimal and $w^{(p)}$ via variational inference [26]. Additional details can be found in the Appendix.

While prior work demonstrated that MIND MELD is capable of improving upon suboptimal demonstrations, MIND MELD suffers from several key limitations: 1) MIND MELD corrects for suboptimality under-the-hood and does not convey to the demonstrator how best to improve their suboptimal tendencies, and 2) MIND MELD assumes that demonstrators are static (i.e., the way in which they are suboptimal does not change over time). Reciprocal MIND MELD overcomes these limitations by 1) providing actionable robotic feedback to the demonstrator to improve upon the quality of their demonstrations and 2) dynamically updating the estimate of their personalized embedding online via our EPN in order to account for changes in suboptimal tendencies and teaching ability.

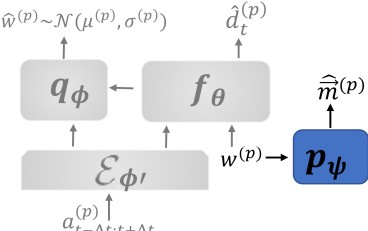

Figure 2: MIND MELD architecture [9] (gray) and additional network head, $p_\psi$, (blue) for learning a semantically meaningful embedding space (see Section 4.1).

**Driving Simulator Domain -** In keeping with prior work [9], we utilize a driving simulator domain based on the high-fidelity physics simulator, Airsim, and an Xbox steering wheel to evaluate Reciprocal MIND MELD. Driving simulators allow researchers to study novel algorithms in an environment that is safe for human subjects. In this domain, participants are tasked with teaching a car to drive from a start location to a goal in various environments while avoiding obstacles. The action space consists of the position of the wheel (-540° to 540°), and the state space consists of images, position, velocity, and acceleration. Feedback is provided to demonstrators via verbal instructions.

## 4   Methodology

Because humans have a greater ability to generalize to novel tasks and domains than a machine-learning algorithm [11], our objective is to provide demonstrators with knowledge about how to improve their demonstrations rather than correcting suboptimality under-the-hood. We propose an approach to reason about a demonstrator's embedding and provide robotic feedback derived from their embedding that is intended to improve upon their demonstration abilities. In keeping with prior work [9], we investigate the abilities of our approach in a driving simulator domain. We break the problem of improving upon a demonstrator's teaching abilities into three research questions.

    **RQ1**: Can robotic feedback improve upon a demonstrator's teaching abilities?
    **RQ2**: What is the best method to provide robotic feedback to improve teaching abilities?
    **RQ3**: Does robotic feedback result in improved learning outcomes on novel tasks and over time?

### 4.1   Semantically Meaningful Embedding Space

Prior work [5, 9] has illustrated that MIND MELD learns embeddings that correlate with suboptimal tendencies and that demonstrators tend to over-/under-correct and provide anticipatory/delayed feedback in a driving simulator domain. We note that domain expertise is required to determine these dimensions of suboptimality. This suboptimality is related to the unintuitive nature of RC LfD as well as the correspondence problem [4, 27] which arises from differences in embodiment between humans and robots. These suboptimal tendencies are unrelated to the specific task itself, but are

related to the task specifications (e.g., providing corrective feedback via a steering wheel). While there may be additional dimensions of suboptimality depending on the robotic domain, we focus our investigation on the over-/under-correcting (o/u) and anticipatory/delayed (a/d) dimensions, as these were determined in prior work to be principle dimensions of suboptimality [9]. We posit that these two dimensions will be common across RC LfD paradigms which require continuous control input and plan to test this hypothesis in future work. Our goal is to learn a *semantically meaningful* embedding space (i.e., a space that can be translated into actionable feedback) and then utilize the location of the demonstrator's embedding within the embedding space to provide robotic feedback.

To learn a semantically meaningful embedding space whose dimensions reflect suboptimal tendencies, we add an additional network head, $p_\psi(w^{(p)}) = \hat{\vec{m}}^{(p)}$, (Fig. 2, shown in blue) to the MIND MELD architecture to estimate the suboptimal tendency, $\vec{m}^{(p)}$, (i.e., the magnitude by which the demonstrator over-/under-corrects and is anticipatory/delayed). We utilize a mean squared error (MSE) loss, $L(\psi, w) = \frac{1}{N} \sum_i \left\| p_\psi(w^{(i)}) - \vec{m}^{(i)} \right\|_2^2$, to train the network to predict the suboptimal tendency, $\vec{m}^{(p)}$, given the personalized embedding. This loss helps to ensure that the dimensions of the embedding space are semantically meaningful and can therefore be translated into actionable robotic feedback. Under IRB approval, we leverage the calibration dataset collected in Schrum et al. [9] to learn a semantically meaningful embedding

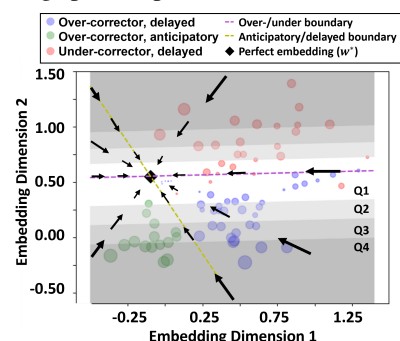

Figure 3: The learned embedding space and decision boundaries. Q1-Q4 indicate quartiles 1-4 for the o/u dimension.

space. This dataset consists of 76 participants who provided demonstrations on a set of calibration tasks. The suboptimal magnitude, $\vec{m}^{(p)}$, is determined via dynamic time warping (DTW) [28] between the participants' feedback and optimal labels from the calibration tasks. Because MIND MELD outputs the difference between the participant's corrective label and the optimal label, the perfect demonstrator's embedding, $w^*$, is defined as the embedding which minimizes the output of the MIND MELD architecture, $w^* = \mathrm{argmin}_{w^{(p)}} \sum_{t,p} f_\theta \left( \mathcal{E}_{\phi'}(a_{(t-\Delta t:t+\Delta t)}^{(p)}), w^{(p)} \right)$, where $a_{t-\Delta t:t+\Delta t}^{(p)}$ is a sequence of demonstrations.

Our next objective is to determine the semantically meaningful dimensions of the embedding space. We train a support vector machine (SVM) with a linear kernel to learn the decision boundaries which best separate the demonstrators into their respective suboptimal categories (o/u and a/d). The SVM training labels are determined via DTW between the participant labels and the optimal labels from the calibration tasks. We utilize an SVM to learn the decision boundaries so that we can add the additional constraint that the classifier must pass through the point representing the perfect demonstrator, $w^*$. The distance between the embedding and the decision boundary along the suboptimal dimension determines the magnitude by which the demonstrator is suboptimal.

Fig. 3 depicts our embedding space with linear classifiers separating over-correctors from under-correctors and delayed from anticipatory. The size of the point represents the magnitude by which the demonstrator is suboptimal in the o/u dimension as determined by DTW. The plot illustrates that demonstrators who are more suboptimal in o/u (as represented by larger points) are farther from the o/u decision boundary, supporting our hypothesis that distance from the decision boundary can be used to measure the degree of suboptimality. To further support our claim, we apply Spearman's correlation and find that distance from the decision boundary strongly correlates with magnitude of suboptimality in both the o/u ($\rho = .84$, $p < .001$) and in a/d dimensions ($\rho = .93$, $p < .001$).

### 4.2 Robotic Feedback

To determine the feedback the robot should provide, we calculate the distance, $\epsilon$, along the semantically meaningful dimension between the personalized embedding, $w^{(p)}$, and perfect embedding, $w^*$, as shown in Fig. 1c and f. In the driving domain, we are interested in $\epsilon_{o/u}^{(i)}$ and $\epsilon_{a/d}^{(i)}$, which define the distance between the demonstrator's embedding and the hypothetical perfect demonstrator's embedding in the o/u dimension and the a/d dimension respectively after the $i^{th}$ round of feedback. In our framework, the feedback is proportional to the distance from $w^*$.

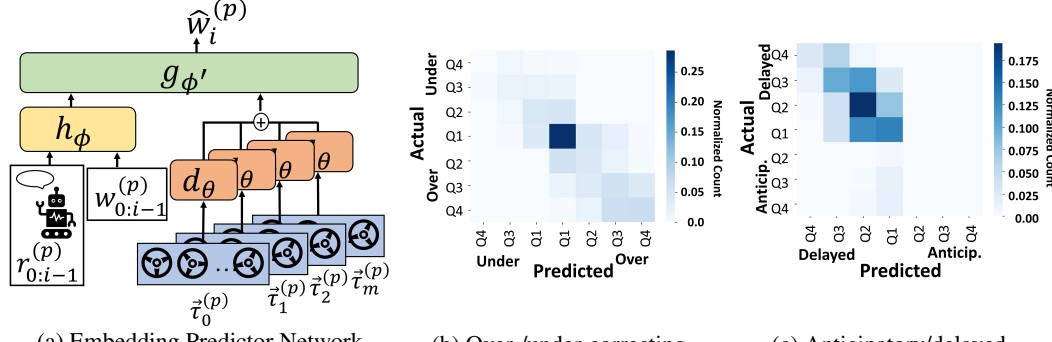

| (a) Embedding Predictor Network | (b) Over-/under-correcting | (c) Anticipatory/delayed |

Figure 4: Fig. 4a illustrates our EPN architecture. Fig. 4b and 4c show the confusion matrices for predicting the quartile that the embedding falls within on holdout test tasks.

To convert $\epsilon_{o/u}^{(i)}$ into actionable and intelligible robotic feedback, we discretize the range of $\epsilon_{o/u}^{(i)}$ by splitting the embeddings from the previously collected calibration participants into quartiles as shown in Fig. 3. Our objective is to move a participant's embedding so that they are in the range denoting the 25% of calibration participants who are the least suboptimal (i.e., quartile one). Participants who fall in a quartile farther from the decision boundary receive feedback proportional to their quartile. For example, if a participant falls in the fourth quartile in the o/u dimension, the robot will instruct the participant to turn the wheel a lot less compared to slightly less in the second quartile. A table showing the feedback for each quartile and dimension can be found in Appendix Table 1.

## 4.3 Online Embedding Estimate

To determine if additional feedback should be provided to the demonstrator and if so, the form of the feedback, we must update our estimate of $w^{(p)}$ after each iteration of robotic feedback. One option to update our estimate of the embedding is to have the demonstrator redo the calibration tasks. However, doing so is time consuming and increases the workload of the demonstrator.

Instead, we propose to dynamically update the embedding online using an LSTM-based architecture which extracts salient features from the demonstrations to estimate the personalized embedding rather than relying on calibration tasks which require known, optimal labels. For example, the velocity and magnitude with which the demonstrator turns the steering wheel are two salient features which can inform the estimate of the new embedding. We call this network the Embedding Predictor Network (EPN) (Fig. 4a). The input to the EPN is the set of new demonstrations, $\tau_{0:m}^{(p)}$, the demonstrator's previous embeddings, $w_{0:i-1}^{(p)}$, and the robotic feedback that was previously provided to the demonstrator, $r_{0:i-1}^{(p)}$. The output of the EPN is an estimate of the new personalized embedding, $\hat{w}_i^{(p)}$. This network utilizes two LSTM subnetworks, $h_\phi$ and $d_\theta$, the output of which is then fed into subnetwork, $g_{\phi'}$, made up of linear layers with ReLU activations. The inputs to $h_\phi$ are $w_{0:i-1}^{(p)}$ and $r_{0:i-1}^{(p)}$. Each trajectory, $\tau_t^{(p)}$, is fed into an LSTM subnetwork, $d_\theta$. We then average across the outputs of $d_\theta$ and feed the result into $g_{\phi'}$ which produces our embedding estimate, $w_i^{(p)}$. We choose to average across the outputs of $d_\theta$ so that our network is agnostic to the number of trajectory inputs.

We train our EPN on the data collected in Studies 1 and 2 as described in Section 5. Fig. 4b and 4c show confusion matrices depicting the ability of the network to accurately predict the quartile of suboptimality in the o/u dimension and the a/d dimension respectively on holdout test tasks.

## 5 Human-Subjects Studies, Results, and Discussion

To determine if Reciprocal MIND MELD is able to improve upon a demonstrator's ability to provide high-quality demonstrations, we conduct three human-subjects studies. The objective of Study 1 is to determine if we are able to shift a demonstrator's embedding via verbal robotic feedback in the o/u dimension (RQ1). In Study 2, we investigate if, and how best, we can shift a demonstrator's embedding in two dimensions (RQ2). In Study 3, we determine if 1) robotic feedback derived from

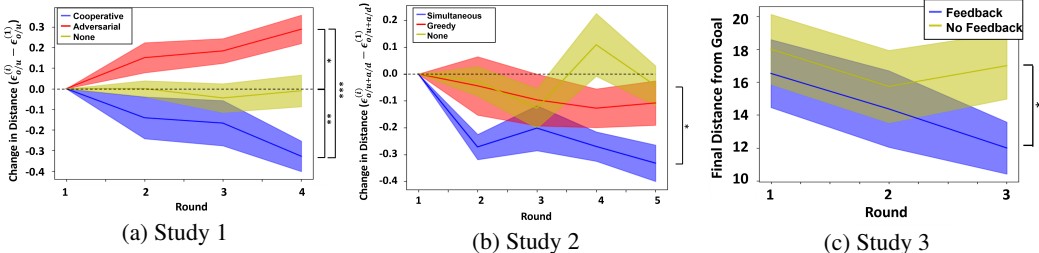

(a) Study 1          (b) Study 2          (c) Study 3

Figure 5: Fig. 5a and 5b show the difference between the embedding distance at round $i$ and the embedding distance at round one for Study 1 and Study 2 respectively. Fig. 5c shows the final distance from the goal for the robot after each round of Study 3.

our EPN rather than the calibration tasks is a good metric of teacher suboptimality and 2) if robotic feedback improves teaching outcomes over time (RQ3). During each study, we employed surveys to measure how robotic feedback altered participants' subjective attitude towards each agent. In our analysis, we check parametric models for normality and homoscedasticity. Model details, tests for assumptions, and additional results are in the Appendix.

## 5.1 Study 1 (RQ1)

Our objective in Study 1 is to demonstrate that robotic feedback can effectively modulate a participant's teaching. In this study, we start by investigating feedback only in the o/u dimension. After completing pre-study surveys, participants complete four rounds of the calibration tasks to measure how their embedding is changing. Participants receive robotic feedback between each round and complete trust [29] and fluency [30] surveys to determine their subjective perceptions of the robot.

**Conditions:** In the *Cooperative* condition, the robot provides feedback to improve the demonstrator's teaching. In the *Adversarial* condition, the robot provides feedback to make the participant a worse demonstrator. In the *None* condition, the participant does not receive any feedback.

**Results:** We recruited 27 participants (Mean age = 24.15, SD = 3.4; 37.0% Female). Fig. 5a shows the change in the distance ($\epsilon_{o/u}^{(i)} - \epsilon_{o/u}^{(1)}$) in the o/u dimension between round one and rounds one through four. We plot $\epsilon_{o/u}^{(i)} - \epsilon_{o/u}^{(1)}$ to show how participants change irrespective of their initial teaching skill. We find that the distance at round one, $\epsilon_{o/u}^{(1)}$, is significantly greater from the distance, $\epsilon_{o/u}^{(4)}$, at round four in Cooperative ($\chi^2(1) = 5.44$, $p = .020$) and significantly less in Adversarial ($F(1,8) = 20.1$, $p = .002$).

We additionally find that Adversarial results in the embedding shifting significantly farther from the perfect embedding between rounds one to four ($F(2,24) = 20.2$, $p < .001$) compared to Cooperative ($p < .001$) and None ($p = .014$). Cooperative shifts the embedding significantly closer to the perfect embedding ($p = .009$) compared to None. Together, these findings indicate that our approach is capable of modulating teaching style in either direction along the suboptimal dimension. Further, the results in None shows that participants are not simply improving due to repeated interactions. Interestingly, we find that participants become significantly worse in the a/d dimension when they only receive feedback in the o/u dimension. We also find that participants' trust increased significantly more ($F(2,24) = 5.15$, $p = .014$) in Cooperative compared to Adversarial ($p = .020$) and None ($p = .038$). Additionally, we find a positive change in fluency ($F(2,24) = 5.10$, $p = .014$) in Cooperative compared to Adversarial ($p = .017$). **Takeaway: Robotic feedback can effectively improve a participant's teaching abilities in a driving simulator domain.**

## 5.2 Study 2 (RQ2)

In Study 2, we next determine how best to provide robotic feedback to both prevent cognitive overload and efficiently improve upon a participant's teaching abilities. Our study design follows the same procedure as Study 1, in which participants complete five rounds of the calibration tasks and receive robotic feedback between each round.

**Conditions:** In *Simultaneous*, the robot provides feedback related to both the o/u and the a/d dimensions. In *Greedy*, the robot only provides feedback related to the condition in which the participant is worst (i.e., farthest from $w^*$). In *None*, the participant receives no feedback.

**Results:** We recruited 39 participants (Mean age = 22.46, SD = 3.3; 38.5% Female). Fig. 5b shows the overall change in the distance ($\epsilon^{(i)}_{o/u+a/d} - \epsilon^{(1)}_{o/u+a/d}$) in the two dimensions of suboptimality between round one and rounds one through five. We find that the distance at round one, $\epsilon^{(1)}_{o/u+a/d}$, is significantly greater from the distance, $\epsilon^{(5)}_{o/u+a/d}$, at round five in Simultaneous ($F(1, 12) = 22.3$), $p < .001$). We next compare $\Delta\epsilon_{o/u+a/d}$ across conditions ($F(2, 36) = 3.77$, $p = .033$). We find that Simultaneous results in the embedding shifting significantly closer to the perfect embedding between rounds one to five compared to None ($p = .034$). We do not find significance between None and Greedy or Simultaneous and Greedy. Participants' trust ($F(2, 36) = 3.81$, $p = .032$) and team fluency ($F(2, 36) = 7.23$, $p = .002$) significantly increased in Simultaneous compared to None ($p = .029$, $p = .002$ respectively). Lastly, although the result is not significant, we find that participant's understanding [31] of the robot increased more in the Simultaneous ($M = 0.61$, $SD = 0.62$) condition compared to None ($M = 0.14$, $SD = 0.69$) and Greedy ($M = .15$, $SD = 0.58$). **Takeaway: Providing feedback in both dimensions simultaneously produces better results for both objective and subjective metrics.**

### 5.3 Study 3 (RQ3)

In Study 3, we aim to show that our approach and the results from Study 1 and 2 translate to improved learning outcomes for an LfD agent on novel tasks. Participants first complete the calibration tasks to obtain an initial estimate of their embedding, $w_0^{(p)}$, and determine $\epsilon^{(0)}_{o/u}$ and $\epsilon^{(0)}_{a/d}$. Next, the robot provides feedback to the participant intended to improve their demonstrations in both the o/u and the a/d dimensions given our positive findings for the *Simultaneous* condition in Study 2. Participants then train the robot for three rounds in three different novel environments (i.e., new start and goal locations) for six demonstrations each. Between each environment, we estimate the participant's new embedding, $w_i^{(p)}$, via the EPN, and calculate $\epsilon^{(i)}_{o/u}$ and $\epsilon^{(i)}_{a/d}$ after each round, $i \in \{1, 2, 3\}$. The robot provides robotic feedback based upon the new estimate of the participant's embedding derived from the EPN. At the end of the study, the participants redo the calibration tasks to determine $\epsilon^{(4)}_{o/u}$ and $\epsilon^{(4)}_{a/d}$. By redoing the calibration tasks, we are able to obtain a ground truth estimate of how the quality of their demonstrations has changed over the course of the study.

**Conditions:** In *Feedback*, the robot provides feedback to the participant about their demonstrations. In *No Feedback*, the robot still interacts with the participant but does not provide feedback.

**Results:** We recruited 60 participants (Mean age = 21.9, SD = 2.89; 28.3% Female). Fig. 5c shows the robot's final distance from the goal for Feedback and No Feedback for rounds 1-3. Participants in Feedback achieve a lower final distance to the goal in round one despite starting off as worse demonstrators on average, as measured via the initial calibration tasks (Mean $\epsilon^{(0)}_{o/u+a/d}$ in Feedback: 0.93, Mean $\epsilon^{(0)}_{o/u+a/d}$ in No Feedback: 0.89). Additionally, the final distance of the robot improves over the rounds in Feedback whereas in No Feedback, the robot improves slightly then gets worse in the final round. In round three, the robot achieves a significantly lower final distance from the goal ($Z = -2.0$, $p = .045$) compared to No Feedback.

To determine if the embedding as estimated by the EPN is a good metric of performance, we compute the correlation between the distance, $\epsilon^{(i)}_{o/u+a/d}$, of the estimated embedding from the perfect embedding and performance, as measured by the average distance from the goal for each round, $i$. Fig. 6 shows a significant correlation ($\rho = .23, p = .002$) between embedding distance and performance, suggesting that the embedding estimated by the EPN is a good measure of suboptimality.

Next, we investigate the overall change in the quality of the participants' demonstrations as measured via the first set of calibration tasks (conducted at the beginning) and last set (conducted at the end). Fig. 7 shows the change, $\epsilon^{(0)} - \epsilon^{(4)}$, for the o/u dimension and the a/d dimension. We find that participants became significantly better in Feedback ($t(52) = 2.62$, $p = .006$) compared to No Feedback in the o/u dimension. While we do not find significance in a/d, we do find that participants

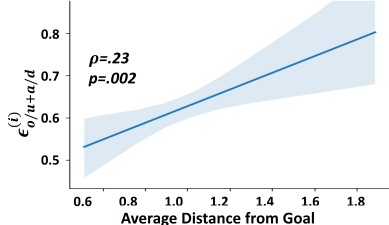

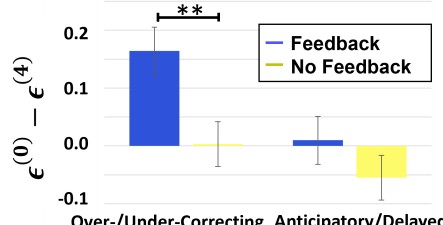

Figure 6: Correlation between the embedding distance, $\epsilon_{o/u+a/d}^{(i)}$, and distance from the goal.

Figure 7: Change in $\epsilon_{o/u}$ and $\epsilon_{a/d}$ between first and last calibration tasks.

improve in Feedback whereas they become worse in No Feedback in this dimension. Lastly, we investigate Feedback versus No Feedback in terms of subjective metrics. Feedback significantly increases trust ($Z = -2.34$, $p = .019$) and decreases workload [32] ($t(58.0) = -1.79$, $p = .039$) compared to No Feedback. **Takeaway: Feedback derived from our EPN improved participant teaching and resulted in better learning outcomes for the robot in novel tasks.**

### 5.4 Discussion and Limitations

In Study 1, we demonstrated we can shift a demonstrator's embedding both farther from and closer to the perfect embedding depending on whether the demonstrator received feedback from an Adversarial or a Cooperative robot respectively ($p < .001$) (RQ1). In Study 2, we found that providing feedback intended to improve upon both dimensions of suboptimality simultaneously is the best strategy and does not cause participants to suffer from an undue level of cognitive overload ($p < .001$) (RQ2). Studies 1 and 2 present strong evidence that robotic feedback is capable of improving upon demonstration quality, suggesting that a robot will learn better from a teacher who has received robotic feedback. In Study 3, we test this hypothesis and investigate the abilities of our EPN to update our estimate of the participant's embedding during novel tasks. We found that final distance of the robot from the goal improves as the demonstrator receives more feedback about their demonstrations ($p = .045$). Overall, we demonstrated that robotic feedback derived from our Reciprocal MIND MELD architecture results in better learning outcomes for a robot in a driving simulator domain.

**Limitations:** A limitation of Reciprocal MIND MELD is that domain knowledge is required to determine the dimensions of suboptimality. However, robotic domains share many similarities in terms of the control interfaces and the potential for suboptimality, suggesting that the dimensions in one domain will likely be similar in others. In this work, we investigate verbal feedback to improve upon demonstration quality. However, prior work has suggested that alternative methods of providing feedback may be more effective at improving teaching abilities [6]. We leave to future work an investigation of the best modality for providing demonstrator feedback in the context of Reciprocal MIND MELD. Furthermore, in this work, we only investigate two dimensions of suboptimality in a driving simulator domain. We plan to investigate, in future work, Reciprocal MIND MELD's ability to generalize to additional dimensions of suboptimality in other domains. Lastly, our population consisted mostly of college aged students. In future work, we propose to sample from a more diverse participant pool.

## 6 Conclusion

We introduce Reciprocal MIND MELD, a novel LfD framework for providing robotic feedback to a human demonstrator based upon a personalized embedding to improve suboptimal teaching tendencies. We demonstrate our approach in a series of three human-subject experiments in which we show that robotic feedback can improve upon the quality of a teacher's demonstrations, providing feedback in multiple dimensions simultaneously is the most effective method, and robotic feedback results in improved learning outcomes for a robot. Additionally, we show that our Embedding Predictor Network is capable of accurately estimating the updated personalized embedding online, thus enabling continuous feedback to be provided to the demonstrator.

# 7 Acknowledgements

This work was supported by the Georgia Institute of Technology State Funding, a NASA Early Career Fellowship under grant 80HQTR19NOA01-19ECF-B1, MIT Lincoln Laboratory under grant FA8702-15-D-0001, a gift from Konica Minolta, and the National Science Foundation under grants 1545287 and 20-604.

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
