# OpenReview forum: "Reciprocal MIND MELD: Improving Learning From Demonstration via Personalized, Reciprocal Teaching"
_robot-learning.org/CoRL/2022/Conference — CoRL 2022 Poster_

### Official Review · Reviewer_tR3N · 2022-07-29

**Originality:** Good
**Technical Quality:** Very Good
**Clarity Of Presentation:** Very Good
**Impact:** 4

**Recommendation:**

Strong Accept: I recommend accepting the paper and will argue for my recommendation even if other reviewers hold a different opinion.

**Summary:**

Authors propose an extension to MIND MELD which aims at improving the human's demonstrations via tailored feedback as opposed to optimising suboptimal demonstrations after the fact. They provide an application specific method for achieving this which is validated extensively by carrying out several experiments with a large number of human participants.

**Issues:**

Too much reliance on appendices and prior work. Should have minimum to understand content in main document. For example, the additional network head in 3.1 is difficult to understand without prior knowledge of MIND MELD. More details on/ example of a calibration task would be helpful. In section 4, at least a high level description of the robot task should be provided as well as how feedback is given. Related worked should be in main document.

**Quality Of The Limitations Section:**

Limitations are addressed clearly

**Reviewer Expertise:**

4: The reviewer is confident but not absolutely certain that the evaluation is correct

**Robotics Focus:**

Highly relevant to robotics but no hardware experiments

**Strengths And Weaknesses:**

Strengths:
- Innovative idea to provide automatic tailored feedback and focus on improving demonstrations which is a largely unaddressed problem in the literature.
- Method is thoroughly evaluated using a large number of human participants and well justified by addressing each of the research questions with focused experiments.
- Paper overall well written and easy to read.
- EPN is interesting and could be applied to other problems.

Weaknesses:
- Relatively simple extension to existing work.
- Meaningful embeddings may not always be simple mappings such as over/under steering. Mapping between corrective actions and meaningful embeddings is simply proportional to the control error. Method is very application specific.

**Summary Of Recommendation:**

Although the proposed method is application specific, the main value in the paper is validating that providing tailored feedback to the demonstrator improves learning performance. This is valuable to the learning from demonstration community since it rationalises a promising direction for future research.

---

> ### Author Response · Authors · 2022-08-20
> **Response to Reviewer tR3N**
>
> We thank the reviewer for their helpful feedback and comments about the work being innovative and thoroughly evaluated on a large number of human subjects. We have made all changes to the text discussed below in red.
>
> * Simple extension to prior work
> >While our approach is based upon a framework presented in prior work, we believe that our insight into how to create a semantically meaningful embedding space that can be translated in actionable robotic feedback, our EPN for predicting suboptimality on-the-fly, and our extensive human subject studies are substantial contributions in relation to prior work. Furthermore, our work is the first to propose a learning approach for providing feedback to improve suboptimal demonstrations.
>
> * Application specific and meaningful embeddings may not be simple mappings
> >While we only demonstrate our framework in a driving application, we hypothesize that the tendency to over-/under-correct and provide delayed/anticipatory feedback will occur for most tasks involving a continuous control input (i.e., wheel, joystick etc.) and may also be present for kinesthetic teaching. Because we do not test these hypotheses, we have included a short discussion in the limitations section (lines 325-328). We agree with the reviewer’s point that complex suboptimal dimensions may be difficult to translate into actionable robotic feedback.
>
> * Too much reliance on Appendix and prior work
> >We appreciate the feedback from the reviewer and have worked to address this concern. We have moved related works (lines 63-76), a more detailed description of the MIND MELD architecture (lines 83-85), a figure depicting the MIND MELD architecture with our new network head (Fig. 2), and details of the driving simulator domain (Lines 106-112) into the main paper. To save space and reduce the dryness of the statistical analysis, we have moved the analysis’ minutiae to the Appendix.
>
> * High level description of task and feedback should be given
> >We added a section to Preliminaries describing the domain and task and how feedback is provided to the demonstrator (lines 106-112).

---

> > ### Author Response · Authors · 2022-08-25
> > **Follow up to Reviewer tR3N**
> >
> > We wanted to thank the reviewer again for their review and to see if there are any additional changes that they would like to see to further improve our paper.

---

### Official Review · Reviewer_oD66 · 2022-07-29

**Originality:** Good
**Technical Quality:** Very Good
**Clarity Of Presentation:** Fair
**Impact:** 3

**Recommendation:**

Weak Accept: I recommend accepting the paper, but will not argue for my recommendation if the majority of other reviewers have a different opinion.

**Summary:**

This work builds on the MIND MELD framework, which improves human-in-the-loop robot learning by having the robot directly model and compensate for the ways that a user's corrections of robot behavior are sub-optimal along two pre-defined dimensions.  In this work, the authors show that telling a user about the dimension of their suboptimality, with feedback tied to their level of suboptimality, can change the user's feedback to be closer to optimal and that the MIND MELD architecture can be extended to compensate for those changes.

**Issues:**

I would love to see a revision that is a little more accurate in its claims -- for example, contribution 3 in the introduction should read: "We validate Reciprocal MIND MELD in a simulated driving task and show that it can improve demonstrators' performance in the anticipatory/delayed dimension of suboptimality [...]"

The limitations section could speak more accurately to the potential for generalization (or not).  For example, in the general case, there is a chicken-and-egg problem with determining suboptimality of demonstrations: the learner needs to know what it looks like to do the task correctly in order to know whether the demonstration is good, but if you have a robust model of a good demonstration, you have a robust model of the task, in which case you don't actually need to learn.


**Quality Of The Limitations Section:**

Additional details required

**Reviewer Expertise:**

5: The reviewer is absolutely certain that the evaluation is correct and very familiar with the relevant literature

**Robotics Focus:**

Highly relevant to robotics but no hardware experiments

**Strengths And Weaknesses:**

Strengths:
- This paper shows that users can be trained to give better feedback, which is likely to be of interest to human-in-the-loop learning researchers
- The authors present three studies to support each component of their work
- The paper provides an example of how a learning algorithm can compensate for changing feedback, which is an important problem in human-in-the-loop learning

Weaknesses:
- The paper is very difficult to understand on its own; too much material has been cut and moved to the appendix (perhaps from a longer submission?).  For example, the task and evaluation environment really do need to be described in the paper, not just the appendix.
- The contributions of the work are over-stated relative to what was actually done (the results are relevant to robotics but no real robot was involved; the "novel" tasks are the same task with a new goal location; it's not clear whether the "robotic" feedback was perceived as coming from the robot; the results are described as applying to suboptimality in general but only two dimensions of suboptimality are studied, and those dimensions are very task-specific; the authors state that feedback improved participant teaching, even though it only improved in one dimension, suggesting that there might be generalization problems)
- The paper does not provide strong evidence that the methods described will generalize to frameworks other than MIND MELD or tasks other than simulated driving.


Questions:
- How are the ground truth labels for the SVM classifier of suboptimality obtained? Why does the proposed method result in semantically meaningful labels? Would this method still result in semantically meaningful labels in another higher-dimensional task?
- What are the calibration tasks?
- How is feedback "provided" to users? Audio? Text?
- How were participants recruited? Why is the gender balance so off for all of the studies?


**Summary Of Recommendation:**

Overall, this paper provides a minor contribution to the field of human-in-the-loop learning with a robotics-relevant validation, but no demonstration on an actual robot.  The most interesting result here is that feedback from a learning system can shift the distribution of demonstrations/corrections even over the course of a single interaction, and that the demonstrations can be made "better" from the robot's perspective.  Many of the technical results are an extension to an existing framework; it's not clear how they would apply to other learning algorithms or frameworks.  I'm a little concerned about how much the authors over-generalize their claims throughout the paper -- for example, "we demonstrate that robotic feedback can effectively improve human demonstrations (p < .001)" (13-14) and "We demonstrate that Reciprocal MIND MELD can improve an individual’s demonstrations across multiple dimensions of suboptimality" (58-59) -- but in all three studies improvements are only shown for the over-/under-correcting dimension, not the the anticipatory/delayed dimension.

---

> ### Author Response · Authors · 2022-08-20
> **Response to Reviewer oD66**
>
> We thank the reviewer for their helpful feedback and insightful questions. We have made all changes to the text discussed below in red.
>
> * Too much material in Appendix
> >We appreciate the feedback from the reviewer and have worked to address this concern. We have moved related works (lines 63-76), a more detailed description of the MIND MELD architecture (lines 83-85), a figure depicting the MIND MELD architecture with our new network head (Fig. 2), and details of the driving simulator domain (Lines 106-112) into the main paper. To save space and reduce the dryness of the statistical analysis, we have moved the analysis’ minutiae to the Appendix.
>
> * The results are relevant to robotics but no real robot.
> >We appreciate the reviewer pointing this out. However, driving simulators are a field standard for evaluation because autonomous vehicles are currently unsafe for university-based human-subjects and machine learning research. For example, Toyota Research Institute conducts much of their human-subjects research for autonomous vehicles with a driving simulator rather than with their actual autonomous vehicles.
>
> * Improvements are only shown in the o/u dimension
> >We respectfully assert that there is improvement in both dimensions. We demonstrate that participants become much worse in the a/d dimension when only receiving feedback in the o/u dimension. However, when we provide feedback in both dimensions participants improve in both dimensions, resulting in a large relative improvement in the a/d dimension (lines 242-244).
>
> * Contributions are overstated
> >We have altered the text in the limitations section (lines 325-328) and explicitly stated throughout the text that we demonstrated our approach only in a driving sim domain and for two dimensions of suboptimality.
>
> * Unclear how method will generalize
> >We fully agree with the reviewer that our framework may not work in all cases. However, we show strong empirical evidence in three human subject studies with 129 participants that our framework can improve suboptimality in our chosen domain.  In future work, we plan to demonstrate the abilities of Reciprocal MIND MELD on a physical robot arm and investigate other dimensions of suboptimality. To address the reviewer’s concerns, we have included a short discussion in the limitations section (Lines 325-328) and have added the qualification that we have only demonstrated in a driving simulator domain. Additionally, we do not claim that our work generalizes to frameworks other than MIND MELD as suggested by the reviewer.
>
>
>
> * Why does the proposed method result in semantically meaningful labels?
> >The labels are semantically meaningful because the dimensions of suboptimality we calculated from the dynamic time warping directly translates to actionable feedback by non-roboticist end-users (over/under-correcting, anticipatory/delayed). Our framework is agnostic to the number of dimensions of suboptimality and therefore can be extended to additional dimensions. To do so, an additional loss can be added when predicting the way in which a demonstrator is suboptimal via their personalized embedding thereby making the embedding space semantically meaningful in the new dimension of suboptimality. In our current framework, we utilize a 2-dimensional embedding space as we found this produced the best results. However, for more complex domains in which there are additional dimensions of suboptimality, a higher-dimensional embedding space may be necessary to capture these dimensions.
>
> * How are the optimal labels for SVM obtained?
> >We revise the text (lines 163-164) to make this more clear.
>
> * What are the calibration tasks?
> >They are a set of tasks meant to be representative of the task space with known optimal labels. We have added additional details to the paper to clarify the calibration tasks (lines 35-36) and added examples to the Appendix.
>
> * How is feedback "provided" to users?
> >The feedback was auditory.  We have revised the paper to state this more clearly (line 112, 215).
>
> * How were participants recruited?
> >Advertising through email lists on our university campus. As we mentioned in our limitations, we would like to sample from a more diverse population pool in the future (lines 328-329).
>
> * Chicken-and-egg problem
> > In our framework, the notion of suboptimality is unrelated to the specific task itself, but is related to the task specifications (e.g., providing corrective feedback via a steering wheel). Therefore, the robot can have a notion of the way in which an individual is suboptimal without knowing how to do a specific task (e.g., drive from point a to point b). By having a set of calibration tasks for which the optimal labels are known, we can determine the way in which the demonstrator is suboptimal on these tasks and then use this information to provide feedback to improve their teaching on future novel tasks in which the optimal labels are not known.  We have clarified this in Lines 127-133.

---

> > ### Author Response · Authors · 2022-08-25
> > **Follow up to Reviewer oD66**
> >
> > We want to thank the reviewer again for their helpful feedback. We are hoping the reviewer might please let us know if our revisions are responsive enough kindly to increase the reviewer's score. If there is anything else we can do to improve the paper, please let us know!

---

> > > ### Comment · Reviewer_oD66 · 2022-08-25
> > > **RE: Author Rebuttal**
> > >
> > > The authors have addressed some of my concerns in the revision (I particularly appreciate the additional information added to the paper about the task and context that helps to make sense of the approach). I think the fundamental problem of generalization/impact remains.  The approach to me is a little stuck in the context of MIND MELD + driving, and future work in this vein (e.g., training users to be better teachers) has the potential to be much more exciting -- if the authors can think more about how their work might speak more to the broader community of researchers in robot learning and human-robot interaction.  This remains a weak accept for me.

---

> > > > ### Author Response · Authors · 2022-08-26
> > > > **Response to Reviewer oD66**
> > > >
> > > > We thank the reviewer for responding to our follow up! We have addressed their additional concerns about generalizability and how our work can speak to the broader community by adding a future works section (below). We hope that the reviewer will reconsider their score in light of these changes. Additionally, we note that we constrained the scope of the paper to a driving simulator domain so that we could thoroughly validate our approach with an accessible, low-cost experimental setup. We systematically conducted three separate human-subject studies with a total of 126 participants to demonstrate the abilities of our framework to improve upon suboptimal teaching.  Conducting multiple experiments in different domains and at this scale would be outside of the scope of a single paper.
> > > >
> > > > Proposed addition to paper: “While we demonstrated Reciprocal MIND MELD in a driving simulator domain, we posit that Reciprocal MIND MELD has the ability to train end-users to be better teachers in many robotic domains. In future work, we aim to investigate Reciprocal MIND MELD with a 7-DOF JACO robot and a set of manipulation tasks for at-home assistive tasks and chores. We will test whether the two dimensions of suboptimality characterized in this paper translate to robot manipulation tasks. Additionally, the ability to learn about the way in which an end-user is suboptimal and communicate feedback to the user applies to many domains beyond LfD to broader human-robot interaction settings. Reciprocal MIND MELD could have potential to be used as a tutoring system to improve human task performance. We propose to explore this potential in future work.”

---

### Official Review · Reviewer_QQMM · 2022-07-30

**Originality:** Good
**Technical Quality:** Good
**Clarity Of Presentation:** Very Good
**Impact:** 2

**Recommendation:**

Weak Accept: I recommend accepting the paper, but will not argue for my recommendation if the majority of other reviewers have a different opinion.

**Summary:**

The authors extend a learning from demonstration approach to provide interpretible feedback on the quality of demonstrations. They show that this feedback does indeed impact how users give demonstrations, that it's possible to estimate how this feedback is impacting the quality of subsequent demonstrations online, and ultimately that users can internalize this feedback to give more useful demonstrations in different task variations.

**Issues:**

* Clarify reasoning for the utility of the approach given the requirement for knowledge of optimal labels for calibration tasks inherited from MIND MELD, and further requirement to have knowledge of structure of suboptimality to apply their extension.

* Consider providing visualizations of domains in the main document (as I believe they're important for readers' understanding), but at least as a supplement.

* The work shares some motivations with an older line of work in providing “teaching guidance,” which you should consider relating to. This work provided generic advice to demonstrators/teachers about how best to teach a particular machine learner. An interesting aspect of the teaching guidance formulation is that it doesn’t assume knowledge of the specifics of a domain, rather it leverages heuristics or in some cases optimal teaching algorithms and attempts to give demonstrators generic advice. Results showed that even this level of unpersonalized advice could help people provide better instruction to classifiers and to inverse reinforcement learners.

  * Eliciting good teaching from humans for machine learners. Maya Cakmak, Andrea L. Thomaz. http://dx.doi.org/10.1016/j.artint.2014.08.005
  * Algorithmic and Human Teaching of Sequential Decision Tasks. Maya Cakmak, Manuel Lopes. https://www.aaai.org/ocs/index.php/AAAI/AAAI12/paper/download/4954/5298
  * Machine Teaching: An Inverse Problem to Machine Learning and an Approach Toward Optimal Education. Xiaojin Zhu. https://ojs.aaai.org/index.php/AAAI/article/download/9761/9620

* Line 126: It isn't immediately clear where the "suboptimal categories" come from. It should be explicit that they come from domain expertise of the referenced work.

* Double check references. Many references incomplete or contain wrong links. 6, 12, 14, 22

* Better contextualize and make explicit how the algorithm relates to reciprocal teaching.


**Quality Of The Limitations Section:**

Limitations are addressed clearly

**Reviewer Expertise:**

4: The reviewer is confident but not absolutely certain that the evaluation is correct

**Robotics Focus:**

Highly relevant to robotics but no hardware experiments

**Strengths And Weaknesses:**

Strengths

* The authors take a more user-centered approach than is common in learning from demonstration work. The position that algorithmic improvements which better learn from suboptimal demonstrations may cause misunderstandings in users is interesting and deserves to be carefully considered.

* It’s powerful to be able provide online feedback to demonstrators and update feedback after each demonstration. This is a worthy goal to strive towards, and the careful and well reported user studies give indication for the value that such a capability can offer.

* I found the paper well structured and easy to read. Figure 1 is an especially helpful roadmap for the paper.


Weaknesses

* It’s unclear when or whether the assumption of a structured understanding of suboptimality makes sense in learning from demonstration. In what setting would the template of this approach be applicable?
    * If the user is demonstrating a novel task because the robot does not yet possess the capability to carry out the task and because programming the task is not practical, then the agent does not have access to a notion of optimality. If it did, it would already know how to do the task. If someone provides a corrective label, how could you draw any conclusion about its suboptimality? This is a problem that the authors inherit from the MIND MELD approach which they extend (which, according to the paper, requires “access to a distribution of calibration tasks from which we can obtain the optimal, ground truth labels”), and one that they exacerbate by making the notion of an embedded representation of demonstrator “suboptimality” core to their approach.
    * If the user is demonstrating a task for which the robot is already equipped with the constituent skills because they need the robot to do things according to their preferences, then LfD is not a particularly desirable interface to strive for. It’s more natural to think of the problem as one of task specification, where language is perhaps more suited. “Cups go in this cabinet, glasses over here.”
    * The authors posit that a generic model of demonstrator characteristics may be possible because robot domains share “many similarities in terms of the control interfaces and the potential for suboptimality” (305). Control interfaces being bad and leading to systematic failings in human provided demonstrations suggests alternative, more effective interventions like improving interfaces or providing instruction and training on the interfaces.
    * In essence, there is a fundamental circularity to the assumptions being made which the authors need to clarify and rationalize.

* The domain and tasks used aren’t illustrated outside of one in Figure 1, which makes it harder to assess the flexibility of the method.
    * The authors describe running participants through calibration, then providing them “novel environments (i.e., new start and goal locations)” (252) to evaluate the feedback approach with. Readers would benefit from seeing what the different instances look like.

* Connection to the concept of reciprocal teaching from the education world doesn’t bear out. The purpose of reciprocal teaching is to engage students more deeply in something that they’re learning by having them externalize it. So, if they’re learning reading comprehension, they externalize strategies for comprehension by engaging in dialogues with their peers that materialize comprehension strategies like attempting to predict what’s going to happen. The main point is that this externalization is helpful (because ensures they are following particular strategies and the teacher can listen and intervene). A side effect is that their peers can also learn from them, instead of always learning directly from the teacher.
    * In this paper, the externalization does not directly aide the agent’s learning process. Rather, it (hopefully) changes how the teacher teaches.
    * Reciprocal teaching would be a model where the agent attempts to provide feedback to other learners, then the teacher could look at the demonstrations the agent is giving and infer something about how it needs to be corrected. Perhaps as a side effect, there would be some utility to whatever the other agents learn.
    * As it stands, the proposed method is more like “teacher evaluation” MIND MELD

**Summary Of Recommendation:**

The way the authors have designed their approach appears to make assumptions that limit its application to problems in the conventional LfD setting. The user studies presented are interesting, but the authors do not provide a persuasive case that the way in which they achieved online feedback would have been meaningfully superior to, for instance, Wizard of Oz-ing the feedback for the sole purpose of studying the interaction questions. I hope the authors can clarify the key assumptions and better illustrate the utility of their approach in a revised version of the text.

**Update after rebuttal:**

The authors' substantial textual revisions have made the paper more transparent. With some of the key assumptions and limitation established early in the paper, I think readers will be better able to assess the approach and form assessments of how such an approach might or might not work in other circumstances, so I have updated my recommendation to weak accept.

Additional feedback regarding "Reciprocal Teaching":
With the explicit connection to reciprocal teaching removed from the text, I think the paper is no longer in danger of being misleading in this connection. With the term now only appearing in the title, there is some chance that a reader aware of the term may stumble for a moment. I want to provide more clarification from this perspective so you can assess whether to change the title.

The Sullivan and Brown paper you reference starts out with an example of reciprocal teaching transcribed from a classroom interaction. A student is leading a discussion on a reading passage with their peers, posing questions, summarizing, and otherwise demonstrating that they are using reading comprehension strategies that they were taught. The teacher provides encouragement and intervenes when the leader gets stuck. The way in which "Reciprocal Teaching" is reciprocal is that both parties (students and teachers) engage in teaching. The reason the parallel breaks down here is that the students **teach each other**. They do not teach the teacher strategies for comprehending text, the teacher already knows these. The teacher roams around between groups, listening to hear whether students are actually engaging in the strategies with their peers. The students benefit from seeing their peers model the strategies, but the key is ultimately that, if they aren't getting it, the act of attempting to externalize the strategies and failing provides the teacher the opportunity to give instruction.

In your approach, the "student" (machine learner) instructs the "teacher" (human) how to teach better. This doesn't provide "opportunity to give instruction," it tells the human to teach differently. It's more akin to a teacher evaluation form given half way through a college class. "It's hard to follow your lecture because you put too much on the slides. Maybe you should do the derivations on the board so we can follow the steps".

---

> ### Author Response · Authors · 2022-08-20
> **Response to Reviewer QQMM**
>
> We thank the reviewer for their helpful feedback and their positive comments about our well-reported user studies. We have made all changes to the text discussed below in red.
>
> * In what setting would a structured understanding of demonstrator suboptimality be appropriate? The assumptions are circular. Clarify utility of approach.
> >In this work, we are considering robot-centric LfD in which the demonstrator provides corrective feedback to the robot. We have revised lines 25-29 of the paper to make this clearer. In our framework, the notion of suboptimality is unrelated to the specific task itself, but is related to the task specifications (e.g., providing corrective feedback via a steering wheel). Therefore, the robot can learn the way in which an individual is suboptimal without knowing how to do a specific task (e.g., drive from point a to point b). We have clarified this in Lines 126-136.  LfD is intended to enable novice users with little programming knowledge to teach robots new skills. Therefore, such a structured understanding of suboptimality would be helpful for a novice user who is unaware of how to provide high-quality demonstrations. By having a set of calibration tasks for which the optimal labels are known, we can determine the way in which the demonstrator is suboptimal on these tasks and then use this information to provide feedback to improve their teaching on future novel tasks in which the optimal labels are not known. However, we acknowledge that it may not be feasible to generate a set of calibration tasks to capture suboptimal tendencies for every end-user application.
>
> * “Control interfaces being bad and leading to systematic failings in human provided demonstrations suggests alternative, more effective interventions like improving interfaces or providing instruction and training on the interfaces.”
> >We agree with the reviewer that poorly-designed control interfaces may require instruction and training and impair the ability of the demonstrator to provide high-quality information to the robot. Furthermore, due to the correspondence problem (Alissandrakis 2007), humans perceive and interact with the world in fundamentally different ways from robots which makes it difficult for humans to understand how to provide high-quality demonstrations to a robot even with an intuitive interface. In our work, we seek to directly tackle this challenge by developing a robotic approach to providing feedback to the demonstrator about how to better utilize these interfaces to provide high-quality demonstrations. We do note, however, our interface was a generic steering wheel design for video game use, and the simulator was Airsim physics simulator with Unreal Engine. We have revised Lines 126-136 of the paper to clarify these points as raised by the reviewer.
>
> * Connection to reciprocal teaching does not bear out.
> >We based the name Reciprocal on the following description: "Reciprocal teaching is an instructional activity that takes the form of a dialogue between teachers and students regarding segments of text for the purpose of constructing the meaning of text." (Sullivan 1986 - https://psycnet.apa.org/record/1986-22967-001).  However, we appreciate the reviewer’s point and have removed the citation to teaching and education literature. We humbly invite the reviewer’s response and are open to changing the title of the paper based upon consideration of our rationale for adopting the term, “reciprocal.”
>
> * It is not clear where suboptimal categories come from.
> >We agree with the reviewer that we have not made this clear. Therefore, we have revised the text (lines 127-128) to clearly state that “domain expertise is required to determine these dimensions of suboptimality.”
>
> * A wizard-of-oz approach may be just as good
> >We agree with the reviewer that a Wizard-of-Oz approach could aid in understanding how various types of feedback could be used to reduce demonstrator suboptimality, particularly in cases in which an automated mechanism has not yet been developed. However, in our work, we develop both a novel, automated mechanism and show in a human-subject experiment that our framework can provide actionable robotic feedback that significantly improves teaching abilities in our driving simulator domain.
>
> * Readers would benefit from seeing what the different instances [of tasks] look like.
> >Thank you for the suggestion.  We have added images of the calibration tasks and novel tasks in the Appendix in Lines 125-127 and Fig. 6 and in Fig. 1 in the main paper.
>
> * Work shares similarities to “teaching guidance”
> >We thank the reviewer for these additional references and have included them in the text in Line 72.

---

> > ### Author Response · Authors · 2022-08-25
> > **Follow up to Reviewer QQMM**
> >
> > We want to thank the reviewer again for their helpful feedback. We are hoping the reviewer might please let us know if our revisions are responsive enough kindly to increase the reviewer's score. If there is anything else we can do to improve the paper, please let us know!

---

> > > ### Author Response · Authors · 2022-08-27
> > > **Additional Follow up**
> > >
> > > We greatly appreciate your feedback on our manuscript. We hope the changes to our manuscript and our explanatory comments improved your evaluation of our work. Since the discussion period will end [today], Saturday August 27, we were wondering if you have any remaining questions for us. Please let us know, and we will happily respond

---

### Official Review · Reviewer_w2jn · 2022-08-01

**Originality:** Very Good
**Technical Quality:** Good
**Clarity Of Presentation:** Fair
**Impact:** 2

**Recommendation:**

Weak Accept: I recommend accepting the paper, but will not argue for my recommendation if the majority of other reviewers have a different opinion.

**Summary:**

The paper proposes learning a model for human suboptimality in giving demonstrations that the robot can use to give the person feedback on how to improve their suboptimality. The robot then updates its estimate of the person’s suboptimality and personalizes its feedback based on it with a novel Embedding Predictor Network (LSTM-based model). The paper presents 3 studies that look at whether robot feedback helps users improve the quality of their demonstrations along one or two dimensions of suboptimality, and whether that leads to improved robot learning.


**Issues:**

The writing has considerable flaws that I believe need to be resolved for acceptance. Evaluating the method on only 2 dimensions of suboptimality is underwhelming, so I would like to either see more dimensions or a convincing explanation for why these 2 dimensions can cover a lot of cases of human suboptimality.

**Quality Of The Limitations Section:**

Additional details required

**Reviewer Expertise:**

4: The reviewer is confident but not absolutely certain that the evaluation is correct

**Robotics Focus:**

Highly relevant to robotics but no hardware experiments

**Strengths And Weaknesses:**

I found the studies to be overall interesting and well-conducted, with large sample sizes and rigorous ANOVA tests and post-hoc analyses. The overall framework is also interesting, very relevant to improving robot learning, and well-motivated.

The central weakness of this paper is that the writing is confusing:
- The abstract and introduction are very unclear in terms of contributions. It was not clear until very late that the calibration tasks were a way to extract ground truth, and because of this, it wasn’t clear that the EPN network was meant to substitute for that until very late too (study 3), so for a majority of the introduction and abstract I did not understand what was a contribution vs what was pre-existing from MIND MELD. I would suggest very clearly stating how EPN is explicitly meant to replace the time-consuming calibration, and describe the 3 study hypotheses from the get go.
- The paper places a few key components in the appendix, which interrupts the flow of the paper and makes it not self-contained. Crucial details (like the entire related work section, the network architecture from MIND MELD, the simulator driving domain used for experiments) are omitted and instead made a part of the appendix. One should be able to understand the key contents of the paper without having to look at the appendix.
- As it is currently written, the method section seems too dependent on certain experimental details: the domain is driving, there are exactly 2 dimensions of suboptimality, etc. Because of this, it’s unclear how much this method can generalize beyond driving or beyond these two hand-chosen suboptimalities.
- The metrics for validating the studies were unclear: what was the purpose of looking at participant trust and fluency? Why did we look at correlation as a metric for EPN feedback being a good measurement of personalized embeddings instead of just verifying against collected ground truth calibration values? Relatedly, the description of the results can be a bit dry at times (many ANOVA numbers and p-values), and I would have preferred to have more intuition and explanations of what the results mean.

Smaller comments:
- The newline between Abstract and the contents of the abstract is unusual, as is having a citation in the abstract.
- Abstract: what is a “personalized embedding”? The term hasn’t been introduced prior to here, so it’s confusing what this sentence means.
- “Prior work has shown that robots learn well via robot-centric learning from demonstration (LfD) [2] because the robot can effectively learn how to recover from mistakes” → this sentence doesn’t make sense: how does LfD help the robot recover from mistakes?
- Fig. 1 is difficult to understand without a more descriptive caption.
- Did not understand Fig. 3b and c.
- Why did you choose the over/under-correcting dimension specifically for Study 1? Given Fig. 6 results, it's unclear how Study 1 would have panned out if anticipatory/delayed were studied instead.
- I don’t understand the last paragraph of Study 1: it seems statistically wrong to compare who has the lower p-value and make statements about that; instead, we should directly compare the two delta values (delta_cooperative vs delta_adversarial).


**Summary Of Recommendation:**

I thought the technical work itself was well done and the studies were interesting and well-analyzed, but the writing was very confusing and it's unclear how much this work generalizes beyond the 2 hand-defined dimensions of suboptimality. For this reason, I am learning weak reject, but I am very willing to have my opinion swayed.

---

> ### Author Response · Authors · 2022-08-20
> **Response to Reviewer w2jn**
>
> We thank the reviewer for the helpful feedback and their positive comments about our studies, analysis, and framework! We made all changes to the text discussed below in red.
>
> * Abstract and introduction are unclear for calibration tasks and contributions
> > We have altered the introduction to more clearly state the goal of the calibration tasks (lines 32-37). We have additionally altered the introduction to more clearly state our contributions and differentiate our contributions from those of MIND MELD [9] (lines 42-52).
>
> * Key components in appendix
> > We appreciate the feedback from the reviewer and have worked to address this. We moved related works (lines 63-76), a more detailed description of the MIND MELD (lines 83-85), a figure depicting MIND MELD with our new network head (Fig. 2), and details of the driving domain (Lines 106-112) into the main paper. To save space and reduce the dryness of the analysis, we have moved the analysis’ minutiae to the Appendix.
>
> * Unclear how method can generalize
> > We agree that we have not shown that our framework can generalize across domains and have made this point clearer by adding to the limitations section in Lines 325-328 of the paper. We note that we consider robot-centric LfD (i.e., imitation learning via DAgger [1]), in which the human provides corrective feedback to the robot. The magnitude (over-/under-correcting) and timing (anticipatory/delayed) of the feedback were determined in prior work [9] to be principal dimension of suboptimality for robotic-centric LfD. We hypothesize that these suboptimality dimensions will be present for many domains involving a continuous control input (i.e., wheel, joystick, kinesthetic feedback, etc.). We agree with the reviewer that there may be a task-specific variability to these dimensions as well, which would need to be explored to show generalization.
>
> * Metrics are unclear
> > We collected multiple subjective metrics to determine if receiving feedback from an agent would improve humans’ perception of the agent. Trust and perceived team fluency are both important predictors of an end-user’s willingness to utilize and interact with a robotic system (Lewis et al. 2018). Additionally, we reported the correlation of the embedding distance with agent distance to the goal to show that embedding distance is not only a good metric of suboptimal tendency but also correlates with actual learning outcomes of the agent. We have clarified why we utilized these metrics in the paper (lines 219-221).
>
> * Why did you pick dimension for study 1?
> > We hypothesized that the o/u tendencies would be the most intuitive for participants to understand and correct for. Adjusting magnitude of feedback is easier for participants than adjusting timing of feedback as timing is linked to a participant’s reaction time. Therefore, we chose to study this dimension first to test if our algorithm could improve upon suboptimal tendencies. We additionally wanted to see how a participant’s suboptimality changed in other dimensions when only provided feedback for one dimension. We found that participants became much worse in the anticipatory/delayed dimension, suggesting it is important to provide feedback for both dimensions which we investigated in Study 2.
>
> * It seems statistically wrong to compare who has the lower p-value
> > We did not compare who has the lower p-value and agree that this would not be best practice. To summarize our procedure, we first conduct an ANOVA (or non-parametric equivalent) to determine if there is a main effect between the categories of the independent variable (i.e., condition: Adversarial, Cooperative, and None). If that effect is significant, then we conduct a Tukey post-hoc test, which assesses pairwise comparisons between each of the conditions while controlling for the family-wise error rate. The Tukey post-hoc test determines if the means between conditions are significantly different from one another. The p-values reported in the paper are the results from the Tukey post-hoc and are not compared to each other. We have removed the unclear statistics to make the results clearer and further discussion can be found in the appendix.
>
> * How does LfD help the robot recover from mistakes?
> > In human-centric LfD (e.g., behavioral cloning), the robot only receives demonstrations of how to do the task correctly. Therefore, human-centric LfD approaches suffer from covariate shift issues due to a mismatch between the distribution of states given by the demonstration versus those experienced by the robot when attempting to accomplish the task. In robot-centric LfD, (e.g., DAgger)  the robot instead learns from a human’s corrective feedback signal at each time step as the robot executes the task which allows the robot to learn how to recover from its mistakes. We have clarified this distinction in Lines 25-29.
>
> * Minor Comments
> > Thank you for pointing out these items. We have addressed all other minor comments in the text in red.

---

> > ### Author Response · Authors · 2022-08-25
> > **Follow up to Reviewer w2jn**
> >
> > We want to thank the reviewer again for their helpful feedback. We are hoping the reviewer might please let us know if our revisions are responsive enough kindly to increase the reviewer's score. If there is anything else we can do to improve the paper, please let us know!

---

> > > ### Comment · Reviewer_w2jn · 2022-08-26
> > > **Response**
> > >
> > > I thank the authors for their diligent work in updating the paper and replying to my questions. After some considerations, I think I will update my score to Weak Accept.

---

> > > > ### Author Response · Authors · 2022-08-26
> > > > **Response to w2jn**
> > > >
> > > > We would like to thank the reviewer for taking into consideration our changes and updating their score. We believe their feedback has greatly improved our paper!

---

### Author Response · Authors · 2022-08-20
**Revised main paper and appendix**

**Comment:**

All revisions are marked in red

**Zip File:**

/attachment/c781398a2479777b6ba29a0e73e15f964539a1ac.zip

---

### Meta-Review · Area_Chair_imxj · 2022-08-14

**Recommendation:** Accept (Poster)
**Confidence:** 5

**Metareview:**

### Strengths
+ interesting topic
+ extensive and thorough experiments
+ potentially very useful and powerful

### Weaknesses
- doubts about assumption of a structured understanding of suboptimality and connection to reciprocal teaching
- key components in appendix
- method and experiments are too interwoven, it is not clear how/if the method will generalize
- contributions not accurately represented, and not clear from the start
- questions about only evaluating 2 dimensions of suboptimality
- domain and tasks need to be explained better

### Summary
A promising paper that will need a quite substantial revision. it will be tricky to make this paper work with the CoRL format/page limit (key components in appendix, might fit a longer journal format better)

### After rebuttal and discussion
The rebuttal and revision clarified the assumptions and limitations. The re-write significantly improved the paper, please also see the "Update after rebuttal" in the review of QQMM.



**Best Paper Nomination:**

No